



# *aridec*: an open database of litter mass loss from aridlands worldwide with recommendations on suitable model applications

Agustín Sarquis [1] [2]; Ignacio A. Siebenhart [1]; Amy T. Austin [1] [2]; Carlos A. Sierra [3][4]

[1] Facultad de Agronomía, Universidad de Buenos Aires, Buenos Aires, 1417, Argentina

[2] Instituto de Investigaciones Fisiológicas y Ecológicas Vinculadas a la Agricultura (IFEVA; CONICET-FAUBA), Buenos Aires, 1417, Argentina

[3] Max-Planck-Institut für Biogeochemie, Jena, 07745, Germany

[4] Swedish University of Agricultural Sciences, Uppsala, Sweden

*Correspondence to:* Agustín Sarquis (agusarquis@agro.uba.ar)

**Abstract.** Plant litter decomposition in terrestrial ecosystems involves the physical and chemical breakdown of organic matter. Development of databases is a promising tool for achieving a predictive understanding of organic matter degradation at regional and global scales. In this paper we present *aridec*: a comprehensive open database containing litter-mass loss data from aridlands across the world. We describe in detail the structure of the database and discuss general patterns in the data. Then, we explore what are the most appropriate model structures to integrate with data on litter decomposition from the database by conducting a collinearity analysis. The database includes 184 entries from aridlands across the world, representing a wide range of climates. For the majority of the data gathered in *aridec* it is possible to fit models of litter decomposition that consider initial organic matter as a homogenous reservoir (one pool models), as well as models with two distinct types of organic compounds that decompose at different speeds (two pool models). Moreover, these two carbon pools can either decompose without interaction (parallel models), or with matter transfer from a labile pool to a slow-decomposing pool after transformation (series models). Although most entries in the database can be used to fit these models, we suggest potential users of this database to test identifiability for each individual case as well as the number of degrees of freedom. Other model applications that were not discussed in this publication might also be suitable for use with this database. Lastly, we give some recommendations for future decomposition studies to be potentially added to this database. The extent of the information included in *aridec* in addition to its open-science approach makes it a great platform for future collaborative efforts in the field of aridland biogeochemistry. The *aridec* version 1.0.1 is archived and publicly available at https://doi.org/10.5281/zenodo.6025969 (Sarquis et al., 2022), and the database is managed under a version-controlled system and centrally stored in GitHub (https://github.com/AgustinSarquis/aridec, last access: 9 February 2022).

## 1 Introduction

Plant litter decomposition has a central role in the balance between carbon (C) storage and losses in terrestrial ecosystems. This process involves the physical and chemical breakdown of organic matter. Together with soil organic matter decomposition, this process is the main route of carbon dioxide ($CO_2$) efflux to the atmosphere in terrestrial ecosystems (Chapin et al., 2011). It also plays a key role in the formation and stabilization of soil organic carbon (SOC; Cotrufo et al., 2013). Therefore, in the context of current global change, a thorough understanding of decomposition is crucial for future C budget and storage predictions (Davidson and Janssens, 2006).

Arid ecosystems (hereafter aridlands) are variously defined as water-limited ecosystems, where the scarcity and unpredictability of precipitation drive most processes (Noy-Meir, 1973). They are also defined as regions where evaporation is higher than precipitation, which in turn limits ecosystem productivity (Jafari et al., 2018). Moreover, aridlands can be classified based on an Aridity Index as hyper arid, arid, semi-arid, and dry subhumid ecosystems (United Nations Environment Programme, 1997). Around 41% of the global land area are considered as aridlands (Safriel and Adeel, 2005) and these systems are expanding due to global change (Feng and Fu, 2013; Reynolds et al., 2007; Yao et al., 2020). Despite their comparatively low productivity, some aridlands can have a potentially large impact on global $CO_2$ dynamics (Ahlström et al., 2015; Poulter et al., 2014). The wide extent of aridland cover and their influence on regional and global biogeochemical cycles make the study of aridlands a priority.

In particular, plant litter decomposition in aridlands is still not well understood (Austin, 2011). Litter mass loss in the field is mainly studied using the litterbag method or some variant (Harmon et al., 1999). The vast majority of litterbag studies come from temperate forests favored by the ease of litter collection and the concentration of researchers close to these study sites. There are fewer studies in aridlands, and few efforts have been made towards synthesizing aridland decomposition literature (Austin, 2011; Cepeda-Pizarro, 1993) or to examine patterns of decomposition in global aridlands (Zhang and Wang, 2015). Nonetheless, substantive literature has already been produced, which would allow for the compilation of a comprehensive database on plant litter decomposition in aridlands that could help boost our understanding of these ecosystems.

Development of databases is a promising tool for achieving a predictive understanding of organic matter degradation at regional and global scales (Luo et al., 2016). This predictive understanding can be obtained through mathematical models,



but there is substantial uncertainty with respect to which models to use. For litter decomposition, some efforts have been made, by fitting models with multiple C pools of different quality that decompose at different rates (Adair et al., 2008), as well as incorporating the effect of abiotic stressors like photodegradation on C dynamics (Adair et al., 2017; Chen et al., 2016; Foereid et al., 2011). Taken together, increased data availability and global representativity of well-constructed databases with our current most complex modeling tools could help us achieve a better understanding of the land C cycle with a higher predictive power.

Once a database of observations has been constructed, there exists the possibility of fitting complex models from these data, although this should be approached with caution. A common issue with mechanistic models used in environmental sciences is that they are poorly identifiable (Brun et al., 2001), meaning that different parameter sets of a model generate similar probability distributions for the observed data (Sierra et al., 2015). In other words, it is impossible to identify a unique set of parameters that explains model behavior. One reason behind this issue is that the information one would like to learn from models is often of a much higher complexity than the information content of the observed data (Brun et al., 2001). It is possible to detect identifiability issues by carrying out collinearity analyses (Sierra et al., 2015; Soetaert and Petzoldt, 2010), among other techniques. Thus, in addition to applying current ecological knowledge about underlying mechanistic processes in model construction, it is important to avoid identifiability problems when fitting these models with real data.

Another important aspect when developing this type of database is to follow an Open Science approach (Hampton et al., 2015). Open Science entails the practice of making all stages of scientific knowledge freely available and presented in a transparent and reproducible way for the whole scientific community to use. Such an approach has the potential to enhance the quality of research products and to speed up scientific progress through collaborative work. Particularly, the development of databases can benefit greatly from an open science perspective by allowing self-motivated reviewers to make comments and by allowing scientists from outside of the core research group to make their own contributions to the database, among other benefits. This latter aspect is key to ensure databases stay updated as new studies get published.

In this paper we present *aridec*: a comprehensive open-science database that comprises time series of litter mass loss (decomposition) data from aridlands across the globe. First, we describe in detail the structure of the database and discuss general patterns in the data. Second, we run a collinearity analysis on the database to explore what might be the most appropriate model structures to fit. We chose a group of models of organic matter loss provided in the R package SoilR as potential models, including models of one, two and three pools with and without matter transfers between them (Sierra et al., 2012). Third, we present an example of applied usage of the database. Lastly, we discuss the scope of the database and give outlines on good field decomposition experimental practices stemming from this work.

## 2 Methods

### 2.1 Database description

We conducted a Scopus search on February 17th, 2021, for field decomposition studies of all times from aridlands published in peer-reviewed journals. We used the search words 'arid OR "dry season" AND decomposition' and excluded results from unrelated subjects. This search produced a list of 1142 publications. To be included in the database, studies additionally needed to fulfil certain criteria: a) field studies in which leaf, shoot or root litter of terrestrial plants was used, and b) minimum of three time points of mass loss data. We did not include wood or dung decomposition studies. We also included publications from our personal libraries. In total, this left us with a list of 184 eligible publications.

We named the database *aridec* and uploaded it to a repository in GitHub (GitHub, 2022; Sarquis et al., 2022). From each selected publication, we created a database entry consisting of three separate files: a file containing time series of mass loss (*timeSeries.csv*), a file containing metadata of the study site and the experimental setup (*metadata.yaml)* and a file with relevant information of the initial characteristics of the litter at the beginning of the experiment (*initConditions.csv)*. We saved each entry in an individual folder named after the last name of the first author and the year of publication. If there was more than one paper per author and year, we added lowercase letters to differentiate them (e.g.: Austin2006a and Austin2006b). We included all entries inside the *data* folder. Other folders in the repository include the *Rpkg* folder containing an R package for querying and manipulating the database, a *test* folder with scripts for testing the integrity of the data and the R package, and an additional folder with miscellaneous *scripts* that demonstrate additional functionality. The overall structure of the database is similar to the structure of SIDb (Schädel et al., 2020) a database of soil incubation time series, and contains the following folder structure:

- *aridec*
    - *Rpkg*
    - *test*
    - *scripts*
    - *data*
        - *single entry*
            - *metadata.yaml*
            - *initConditions.csv*
            - *timeSeries.csv*

The *timeSeries.csv* file includes litter mass loss over time as reported in the original publication. It is a csv type file ("comma-separated values") with column names in the first row. The first column name is always the variable Time and the first value in this column is always "0" (zero). Successive time values should be specified according to each sampling date reported in the study. Time units accepted are days, weeks, months, and years. Starting from the second column, column names should be unique variable identifiers. Below these names, mass loss data should be included as a percentage

of the initial value, which is always 100. When data in the paper are reported in graph form, it is necessary to extract data point values with software tools such as WebPlotDigitizer (Rohatgi, 2020). Acceptable mass loss units are percentage of remaining dry weight, dry organic matter, dry ash-free mass, or C. For remaining mass data as well as for time, units should be specified in the *metadata.yaml* file described below.

The *metadata.yaml* file includes additional information reported in the original paper. It is a yaml type file ("YAML ain't

markup language"), which allows us to write lists of items in a hierarchical form and is both machine and human readable. It includes four main sections: entry identification data, the *siteInfo*, the *experimentInfo* and the *variables* sections. A template for this file with a full description of how to complete it is available inside the *data* folder. The first part includes the *citationKey* which is a unique identifier for the whole entry in the format LastnameYEAR (lowercase letters must be added when there are two or more entries by the same author and year, i.e.: LastnameYEARa and LastnameYEARb).

This *citationKey* name should be the exact same as the folder name. Next is the *doi*, which stands for the Digital Object Identifier where data is published. *entryAuthor* and *contactName* are both first and last name of the person who wrote the entry file and their supervisor (only if applicable), respectively. If the entry author works independently of a supervisor, both fields should be filled with the same name. *contactEmail* should be filled with the supervisor's e-mail address. *entryCreationDate* stands for the date when the file was created, following the format: YYYY-MM-DD. *entryNote* should

include any notes or comments related to this entry, such as missing data or additional data sources used to complete this file. Lastly, *study* requires a short study description in not more than one sentence.

The second part of the *metadata.yaml* file is the *siteInfo* section, which includes environmental information of ecological interest from the study site. First is the *site* field that requires an identification name for the site (not necessarily the site's real name). If the study includes more than one site, an array format should be used in this field and the rest of the items

in this section should be arrays of equal length. The *coordinates* field should be completed using decimal units, checking for the negative sign that denotes southern and western hemispheres. If absent from the publication, *coordinates* can be approximately obtained from Google Earth (Google LLC, 2020). The *country* field should be completed avoiding full names (e.g.: "China" instead of "People's Republic of China" or "USA" instead of "United States of America"). Mean annual temperature (° C) and mean annual precipitation (mm) should be entered in the fields *MAT* and *MAP*, respectively.

When climatologic data are absent from the paper, they can be retrieved from other databases like the POWER database [NASA Langley Research Center (LaRC)]. The *rainySeason* field should be filled with either one of five options: whole year, spring, summer, autumn, winter; if precipitation does not follow a unimodal pattern, this item is left blank. Elevation of the study site in m a.s.l. should be entered under the elevation field, which if absent from the publication can be retrieved from other sources such as Google Earth. The type of vegetation cover of the site should be specified in *landCover*, with

possible options: marsh, greenbelt, farmland, mangrove forest, subalpine, shrubland, urban, sandland, forest, steppe, desert, grassland, and savanna. The item *vegNote* should include a short description of not more than one sentence of the species or functional type composition at the site, if available. The *cover* item should be completed with percentage values of total plant cover or with cover values for specific plant functional types, as available. Lastly, the *soilTaxonomy* item must be completed using the taxonomic classification of the soil at the site. If the classification system used in the paper

is unknown, it is better to leave this section blank, for exact equivalences between soil classification systems are unlikely. The third part of the *metadata.yaml* file is the *experimentInfo* section which includes information regarding the experimental design of the study. *incDesc* stands for incubation description and must include a short list of treatments and sampling points in time. The number of replicates should be specified, paying attention to occasional pseudo-replication in decomposition studies. Experiment *duration* in days should be completed with the maximum time length that samples

stayed in the field. The month in which the experiment started should be specified under *startingMonth*. The name of the litter used for the experiment should be specified under *litter*, and it should match the name used in the *initConditions.csv* file (see below). Under the *litterbag* field, many sub-fields for different characteristics of interest should be completed, such as mesh material, mesh size (one side of a square in mm), dimensions (in cm), mesh transmittance (as a percentage of full sunlight) and litterbag position (full list of options available in the template file). A general rule for the

*experimentInfo* section is that when there is more than one option for a field they should be considered as different levels of a treatment. In this case that field should be left blank in this section, and a new field should be created in the *variables* section by replacing the *experimentalTreatment* placeholder in each variable (see below)

The last section of the *metadata.yaml* file is the *variables* section, which serves as a link between columns in the *timeSeries.csv* file and metadata. Thus, this section should have as many variables as columns in the *timeSeries.csv* file.

The first variable (V1) must always be called "Time" and only time units should be modified accordingly. The rest of the variables (V2 to Vn) must be adequately edited to represent treatment application as described in the original publication. Variable names should match column names in the *timeSeries.csv* file. Litter mass loss *units* should be expressed either in (dry) mass remaining, organic matter (or ash-free dry mass) remaining, or C remaining. Under varDesc (as in variable description) one should write a brief sentence indicating specific treatment levels applied to this variable. The *site* field

should be completed using the same site name entered in the *siteInfo* section. The *experimentalTreatment* item is a place holder for treatments with multiple levels. It should be replaced by any of the listed variables in *experimentInfo* and





completed with an appropriate treatment level. In *compTreat* complementary treatments not included in the rest of the
metadata items should be indicated using key words (e.g.: grazed/ungrazed, water addition, control, etc.). Finally,




**Figure 1: A guiding flowchart of the entry-submitting process for potential contributors of *aridec*.**





transmittance and wavelength threshold (nm) data for radiation filters should be indicated under *filter*. This sub-section should be completed only for photodegradation studies.

The last file in the *data* folder is the *initConditions.csv* file which contains details on the plant litter substrate used for each experiment. The first row contains column names. The first column name is *species* and is the only mandatory item, nonetheless we strongly recommend completing all items if possible. We suggest checking for the correctness of scientific names in the Global Biodiversity Information Facility database (GBIF.org, 2022). Names in the *species* column should be used to complete the *litter* item in the *metadata.yaml* file. Four options are valid for the *type* column: deciduous or evergreen (for woody plants) and forb or graminoid (for herbaceous plants). For the *N-fixer* item we recommend

consulting the *NodDB* database (Tedersoo et al., 2018). Units for the sample *amount* column are in g, for the nutrients and fibers in percentage and for *SLA* (specific leaf area) in mm$^2$ mg$^{-1}$. When litter quality traits are not provided in the original paper, they can be obtained upon request from the *TRY* database (Kattge et al., 2020). We created a template for the *initConditions.csv* and a *README.md* file with further instructions in the *data* folder. Special attention should be made to the *material* section of the *README.md* file, for litter substrates are highly variable among studies and this is

key for database consistency. In Fig. 1, we present a flowchart with the full process of entry submitting for potential contributors.

We generated a Global Aridity Index (GAI) map with the study sites from the database. We retrieved GAI data from the Consortium for Spatial Information global climate data sets (CGIAR-CSI; Trabucco & Zomer 2018). This index is calculated after dividing mean annual precipitation by mean annual reference evapotranspiration. The raster data set we

used is based on WorldClim2 database (Fick & Hijmans, 2017). We chose this dataset because it encompasses a relatively long period of time (from 1970 to 2000) and it has a high spatial resolution (~1 km at the Equator). We then classified each study site by its GAI value as hyper-arid (0 - 0.05), arid (0.05 - 0.2), semi-arid (0.2 - 0.5), dry sub-humid (0.5 - 0.65) and humid (0.65<; United Nations Environment Programme, 1997). Complementarily, to explore how representative our sites in *aridec* are of the whole climatic range of aridlands, we first made a random point-sampling of 6793 pixel units

separated at least 1 km away from each other within the range of aridlands (GAI: 0 - 0.65). For each sample, we averaged mean monthly temperatures from WorldClim2 to obtain mean annual temperature values. We did the same for our *aridec* coordinates and plotted both sets of data together to evaluate how well aridlands are represented in our database. We used the QGIS software to process data and create a map (QGIS Development Team, 2021).

**2.2 Model fitting and collinearity analysis**

A central application of this database is the development of models of litter decomposition for aridlands. In an attempt to explore what are the most appropriate model structures to integrate with data from the database, we selected different structures of decomposition models based on recent theory of models of organic matter decomposition (Sierra and Müller, 2015). These model structures are already implemented in the *SoilR* package (Sierra et al., 2012), and we provide here an

interface between our database and this R package. *SoilR* is a modelling framework that contains a wide set of functions and tools to model soil organic matter decomposition within the R computing platform (R Core Team, 2020).

Organic matter decomposition in *SoilR* is represented by systems of linear differential equations that generalize most compartment-based models. A simple general structure to represent litter decay with no inputs follows Equation 1:

$$\frac{dC(t)}{dt} = A\,C(t) \tag{1}$$

$$C(t) = [C_{pool\ 1},\ ...,\ C_{pool\ m}]^{\mathrm{T}}$$

$$A = \begin{bmatrix} -k_1 & \cdots & a_{1i} \\ \vdots & \ddots & \vdots \\ a_{j1} & \cdots & -k_m \end{bmatrix}$$


Where *C(t)* is a *m x 1* vector with *m* pools of litter mass observed at time *t*, and *A* is a square *m x m* matrix that contains decomposition rates ($k_m$) for each pool and transfer rates ($a_{ij}$) between them. These different pools may correspond to different ways in which the quality of the litter is expressed in different studies. For example, they may correspond to different compounds obtained from a specific extraction method (e.g., water soluble sugars, or acid detergent lignin), or

they can be defined by general decay classes such as fast and slow decay compounds. These pools have different decomposition rates, pool 1 being the fastest decomposing pool and pool m being the slowest. The linear dynamical system represented by Eq. (1), has many different solutions, but we are only interested in the solution that satisfies

$$C\,(t = 0) = C_0 = [total\ C_0 \cdot p_1,\ ...,\ total\ C_0 \cdot p_m]^T \tag{2}$$


where $C_0$ is a *m×1* vector with the value of initial litter mass content in the different compartments *m*. Total $C_0$ is set to be 100% in *SoilR* for our database and resulting parameters $p_m$ are the initial proportions of litter in *m* pools. Using this framework, we chose to fit a total of five different models with an increasing number of parameters (Table 1).



| Model structure | m | Parameters |
|---|---|---|
| Two-pool parallel | 2 | $k_1$, $k_2$, and $p_1$ |
| Two-pool series | 2 | $k_1$, $k_2$, $p_1$, and $a_{21}$ |
| Two-pool with feedback | 2 | $k_1$, $k_2$, $p_1$, $a_{21}$, and $a_{12}$ |
| Three-pool parallel | 3 | $k_1$, $k_2$, $k_3$, $p_1$, and $p_2$ |
| Three-pool series | 3 | $k_1$, $k_2$, $k_3$, $p_1$, $p_2$, $a_{21}$, and $a_{32}$ |

**Table 1: fitted model structures and parameters. m: number of C pools. $k_1$, $k_2$, $k_3$: decomposition rates of pools 1, 2 and 3, respectively. $p_1$, $p_2$: initial proportions of C in pools 1 and 2, respectively. $a_{21}$: transfer rate from pool 1 to pool 2. $a_{12}$: transfer rate from pool 2 to pool 1. $a_{32}$: transfer rate from pool 2 to pool 3.**

For this set of models, we performed an identifiability analysis following the procedure described by Soetaert & Petzoldt
(2010). Non-identifiability is a common issue with inverse-modelling approaches. It is a type of model overparameterization that makes precisely determining parameter values virtually impossible, thus parameters are "non-identifiable". When parameters are functionally related, changes in one parameter can be compensated by changes in others. This produces different parameter sets that have similar probability distributions, thus the inability to determine a single parameter set for the model (Sierra et al., 2015). Analyzing for parameter identifiability in models fitted with *aridec* data allowed us to assess which model structures are the most appropriate to use in this context.
This identifiability analysis is based on the calculation of the collinearity index (Brun et al., 2001). This index is a measure of the degree to which changes in one parameter are compensated by changes in other parameters for a certain model structure and data set. We used the *modCost* function from the *FME* R package to first adjust a model cost function (Soetaert and Petzoldt, 2010). This function estimates weighted residuals of the model output versus the observed data and calculates sums of squared residuals, according to the formula:

$$res_{k,l} = \frac{Mod_{k,l} - Obs_{k,l}}{error_{k,l} \cdot n_l} \tag{3}$$

where $Mod_{k,l}$ and $Obs_{k,l}$ are the modeled and observed values for any data point, $k$, of a variable $l$, respectively. $error_{k,l}$ is a weighing factor that makes the term non-dimensional.
The model cost function, together with a set of pre-set initial parameter values, is then used as an input to calculate a matrix of sensitivity functions using the *sensFun* function from *FME*. This function estimates the sensitivity of the model output to the parameter values using the expression:

$$S_{ij} = \frac{\partial r_i}{\partial \Theta_j} \cdot \frac{w_{\Theta_j}}{w_{r_i}} \tag{4}$$


where $S_{i,j}$ represents each entry of the matrix, $r_i$ are model residuals calculated from the cost function, $\Theta_j$ is a model parameter, $w_{r_i}$ is the scaling of $r_i$, and $w_{\Theta_j}$ is the scaling of parameter $\Theta_j$ (Soetaert and Petzoldt, 2010).
The final step in this analysis is calculating the collinearity index $\gamma$. The *collin* function from *FME* uses the sensitivity matrix as an input to calculate $\gamma$ for every combination of parameters. $\gamma$ is defined as

$$\gamma = \frac{1}{\sqrt{min(EV[\hat{S}^T \hat{S}])}} \tag{5}$$

where

$$\hat{S}_{ij} = \frac{S_{ij}}{\sqrt{\Sigma_j S_{ij}^2}} \tag{6}$$

where $\hat{S}_{ij}$ contains the columns of the sensitivity matrix that correspond to the parameters included in the set and $EV$ estimates the eigenvalues. The collinearity index equals 1 if the columns are orthogonal, and the set is identifiable. The index equals infinity if columns in the sensitivity matrix are linearly dependent (Soetaert and Petzoldt, 2010). The
interpretation of the collinearity index is thus: a change in the residuals caused by a change in one of the parameters can be compensated by a proportional change $1/\gamma$ in another parameter. For practical purposes, if $\gamma > 20$ the parameter combination is considered non-identifiable (Sierra et al., 2015).
For the identifiability analysis, we first selected a representative group of 30 entries from the database ranging from 3 to 19 time points (Table 2). The number of data points in time limits the number of parameters that can be fitted because it
affects the number of degrees of freedom. Thus, models with more parameters require longer data sets. This meant that, a priori, not all entries could be used to fit all model structures. On the other hand, it is possible to test identifiability for restricted model versions, that is models with some of their parameters fixed to a known value. This implies there are less parameters to be determined, thus it allows to use shorter time series. The details of all the models tested are reported in Table 3. From this first analysis, we noticed that two pool parallel and series structure models with a restricted initial



proportion of litter in pool 1 ($p_1$) were the two models more likely to meet identifiability with our data. Because of this, we tested collinearity for all the 184 entries in the database, but only for these two models and for the respective models with the full set of parameters, for comparison. R code for this analysis can be found in the *collinearity.R* script inside the *scripts* folder of *aridec*.

| Number of time points | *aridec* Citation Key | Publication DOI or URL |
|---|---|---|
| 3 | Classen2007 | 10.1111/j.1365-2745.2007.01297.x |
| | Correa2016 | 10.3832/ifor1459-008 |
| | Dominguez2010 | 10.1016/j.still.2010.06.008 |
| | Gehrke1995 | 10.2307/3546223 |
| | Gliksman2018a | 10.1111/1365-2435.13018 |
| 4 | Bernaschini2019 | 10.1016/j.jaridenv.2015.11.009 |
| | delCid2019 | 10.1556/168.2019.20.3.10 |
| | Dipman2019 | 10.1016/j.apsoil.2019.07.005 |
| | Glassman2018 | 10.1073/pnas.1811269115 |
| | Henry2008 | 10.1007/s10021-008-9141-4 |
| 5 | Almagro2015 | 10.1016/j.soilbio.2015.08.006 |
| | Almagro2017 | 10.1007/s10021-016-0036-5 |
| | Brandt2010 | 10.1007/s10021-010-9353-2 |
| | Bucher2003 | 10.1017/S0266467403003377 |
| | Campos2017 | 10.1007/s11104-017-3221-1 |
| 6 | Bosco2016 | 10.1007/s11104-016-2864-7 |
| | Canessa2021 | 10.1111/1365-2745.13516 |
| | Chuan2018 | 10.1007/s10021-018-0221-9 |
| | Connin2001 | 10.1016/S0038-0717(00)00113-9 |
| | DiedhiouSall2013 | 10.2136/sssaj2012.0284 |
| 7 | Araujo2012 | 10.1007/s00442-011-2063-4 |
| | Austin2006a | 10.1007/s10021-005-0039-0 |
| | Bates2007 | 10.1016/j.jaridenv.2006.12.015 |
| | Brandt2007 | 10.1111/j.1365-2486.2007.01428.x |
| | Hou2019 | 10.1080/03650340.2019.1639156 |
| 8 | Day2018 | 10.1111/gcb.14438 |
| | SanchezAndres2010 | 10.1016/j.ecss.2010.07.005 |
| | Xie2020 | 10.1002/ece3.6264 |
| 13 | Arriaga2007 | 10.1007/s11258-006-9178-4 |
| 19 | Ilangovan1996 | http://www.jstor.org/stable/43582052 |

**Table 2: *aridec* entries used in the identifiability analysis with their corresponding DOI o URL. The number of time points refers to the number of sampling dates at each study plus the initial date.**

| Model structure | 2-pool models | | | | 3-pool models | | | |
|---|---|---|---|---|---|---|---|---|
| | T | P | D | C | T | P | D | C |
| Parallel | 3 | 3 | 30 | 120 | 6 | 5 | 15 | 390 |
| Series | 4 | 4 | 25 | 275 | 8 | 7 | 5 | 935 |
| Feedback | 6 | 5 | 15 | 390 | - | - | - | - |

**Table 3: Minimum number of time points in data sets fitted to each model structure T, number of parameters for each model structure P, number of data sets used for each model structure D, and possible number of combinations of**
**parameters to identify with specific combinations of available data C.**

### 2.3 Applied Example

Our collinearity analysis (see below) showed that although most entries could be used to fit two-pool parallel and series models with a fixed $p_1$ parameter (i.e., the initial proportion of litter in pool 1), this was dependent on each data set. Restricting the $p_1$ parameter is a sensible way of achieving identifiability because it is common to find information on



litter lignin content in decomposition publications, and this can be used as an initial proportion value for the slow-decomposing litter pool (i.e.: $p_2$). Since

$$p_1 + p_2 = 1, \hspace{6cm} (7)$$

then it is possible to estimate the $p_1$ as the complementary value of $p_2$.

To give a practical example of what can be done with this database, we chose to fit these models using one of the entries where both the models were identifiable, and the initial proportion of litter lignin was available. Together with these models, we fit a simple one pool model for comparison. We used variable 2 (V2) from the *Day2018* entry which corresponds to *Simmondsia chinensis* (Link) C. K. Schneid. leaf litter decomposed under full sunlight treatment in the field (Day et al., 2018). The initial proportion of lignin ($p_2$) was 0.09. We used the Bias Corrected Akaike Information Criteria (AICc) to assess model fit (Shumway and Stoffer, 2017).

## 3    Results

### 3.1   Data overview

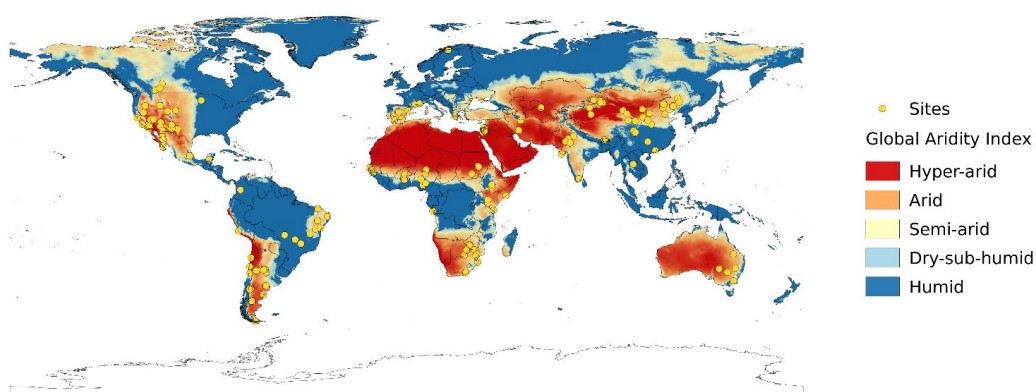

**Figure 2: GAI map generated using data from WorldClim 2 database. Hyper-arid: 0 - 0.03. Arid: 0.03 - 0.2. Semi-arid: 0.2 - 0.5. Dry sub-humid: 0.5 - 0.65. Humid: 0.65<. Green points represent study sites in the *aridec* database.**

   The 184 studies in the database included data for 212 unique study sites around the world. Twenty-four of these sites were repeated in two or more studies. According to the GAI (), ~3.6% of sites were classified as hyper-arid (0 - 0.05), ~35.9%
as arid (0.05 - 0.2), ~43% as semi-arid (0.2 - 0.5), ~6% as dry sub-humid (0.5 - 0.65), and ~12% as humid (0.65<). We recognize humid sites do not classify as aridlands, but we included them nonetheless because these sites had marked dry seasons according to the original publications. A total of 33 countries were represented in our database. Top-five countries with the biggest amount of study sites were China (58 sites), USA (49 sites), Argentina (32 sites), Israel (22 sites) and Brazil (12 sites). Most sites in the database correspond to arid regions were mean annual temperatures are above zero
degrees Celsius, with a very low representativity of colder regions (Fig. 3)
   Out of the 184 database entries, we retrieved 1752 series of litter mass loss over time. The oldest publication in the database is from 1975 and the newest from 2021, and there has been a considerable growth in the number of publications per year (Fig. 4a). Study duration in the database ranged from 18 days to 10 years, with a median of 365 and a mean of 430 days (Fig. 4b). The number of sample harvests from the field went from 2 to 23, with a mean of ~6 and a median of
5 (Fig. 4c). Sampling frequency ranged from 0.08 to 11.1 samples per month, with a median of 0.4 and a mean of 0.8 samples per month (Fig 3d). Elevation at the study sites varied from -375 to 4000 m a.s.l., with median and mean values of 557 and ~811 m a.s.l., respectively (Fig. 4e). Mean annual temperature ranged from -0.45 to 29.5 °C at the study sites with a mean value of 14.9 °C and median of 15.6 °C (Fig. 4f). Mean annual precipitation in *aridec* ranged from 2 to 1700 mm, with median and mean values of ~375 and ~494 mm, respectively (Fig. 4g). Out of all sites, 23 % were reported by
the authors to be deserts, 17 % forests, 16 % agroecosystems, 12% grasslands, 10 % shrublands, 10 % steppes, 8 % savannas, 2 % coastal ecosystems, and 2 % urban sites (Fig. 4h).

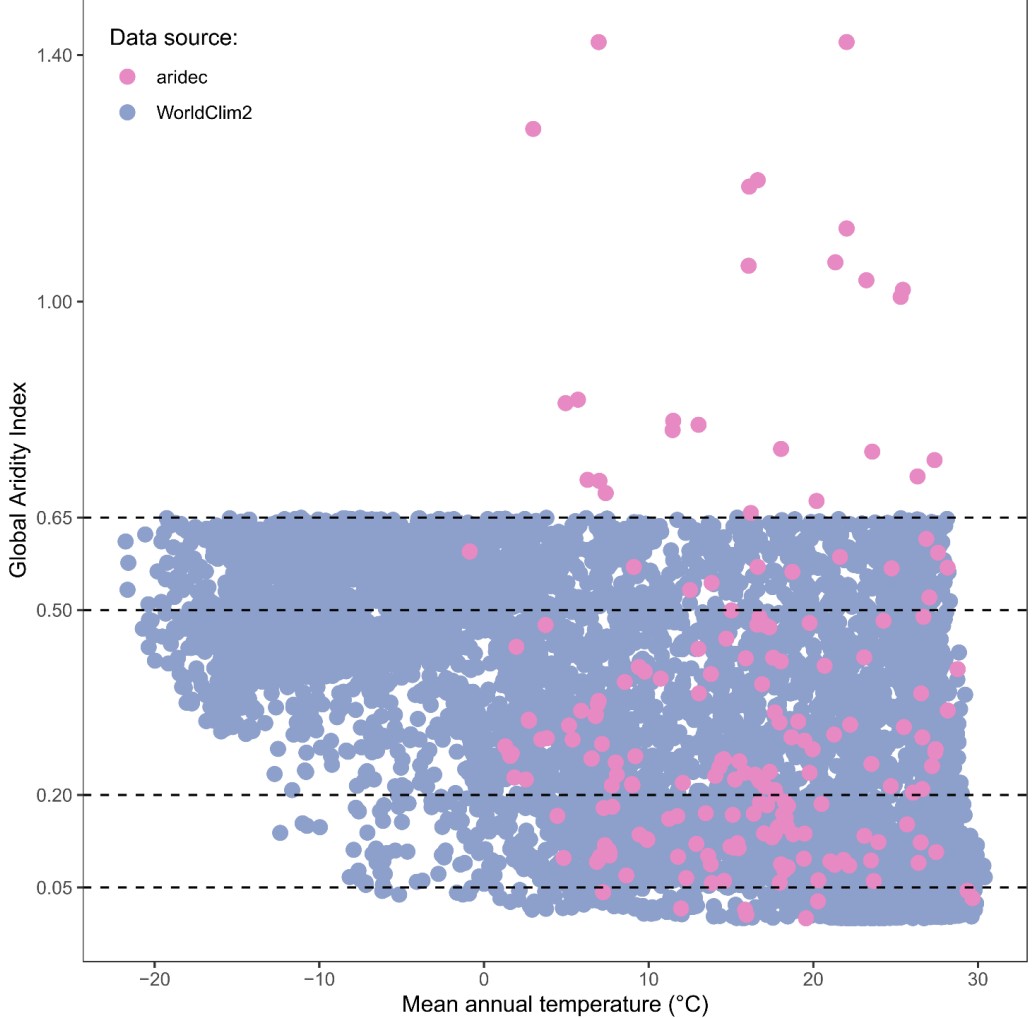

**Figure 3: Climate representativity of the aridec database. GAI plotted against Mean annual temperature (°C). WorldClim2 points were generated via a random sampling of 6793 pixels in QGIS. All data comes from WorldClim2 database. Horizontal dashed lines represent the breaks in GAI between aridland categories: hyper-arid, 0 - 0.03; arid, 0.03 - 0.2; semi-arid, 0.2 - 0.5; dry sub-humid, 0.5 - 0.65; humid, 0.65<.**

### 3.2 Identifiability analysis

Figure 5 shows results from the first identifiability analysis carried out on a subset of 30 entries. For the two-pool parallel model structure four parameter combinations were compared: a full three parameter model and three alternative models with one restricted parameter each. There were 30 points for the full model with 62.1 % of values below the collinearity index $\gamma = 20$ threshold, i.e.: almost 40 % of the models were not identifiable. For the models restricted to two parameters, 95.6 % out of the 90 models compared were identifiable. When specifically looking at the model with a restricted $p_1$ value, 100 % of the models were identifiable. This was expected because usually the fewer the parameters to be estimated, the lower the collinearity in models.

The two-pool series model structure analysis included: a full four parameter model and ten alternative models with one or two restricted parameters each. From a total of 25 full parameter models, 48 % were identifiable according to their $\gamma$ values. Out of 150 models with two restricted parameters, 94.7 % of models were identifiable, while for models with only one restricted parameter, 68.3 % of models were identifiable. When specifically checking for the collinearity index in models with a restricted $p_1$ parameter, 84.6 % of them were identifiable. The non-identifiable values in this last case


**Figure 4:** *aridec* data overview including number of publications per year (a), study duration in days (b), number of sample harvests (c), study sampling frequency as number of samplings per month (d), study site altitude in m a.s.l. (e), mean annual temperature in °C (f), mean annual precipitation in mm (g), and number of studies per type of land cover (h). Dashed lines represent the mean and dotted lines represent the median in each panel.

corresponded to four entries with four time points each. That means that 100% of models with >5 time points were identifiable for the restricted $p_1$ model version. This highlights the importance of having longer time series available for modelling.

The case of the two-pool model structure with feedback included: a full parameter model and 24 other model variants with one, two and three restricted parameters each. The analysis of 15 models with all five parameters showed 100 % of non-identifiable results. The results for the restricted model version with four parameters showed 100 % of non-

identifiable models out of 75 data points, while only 4.7 % of the 150 data points with three parameters gave $\gamma$ values

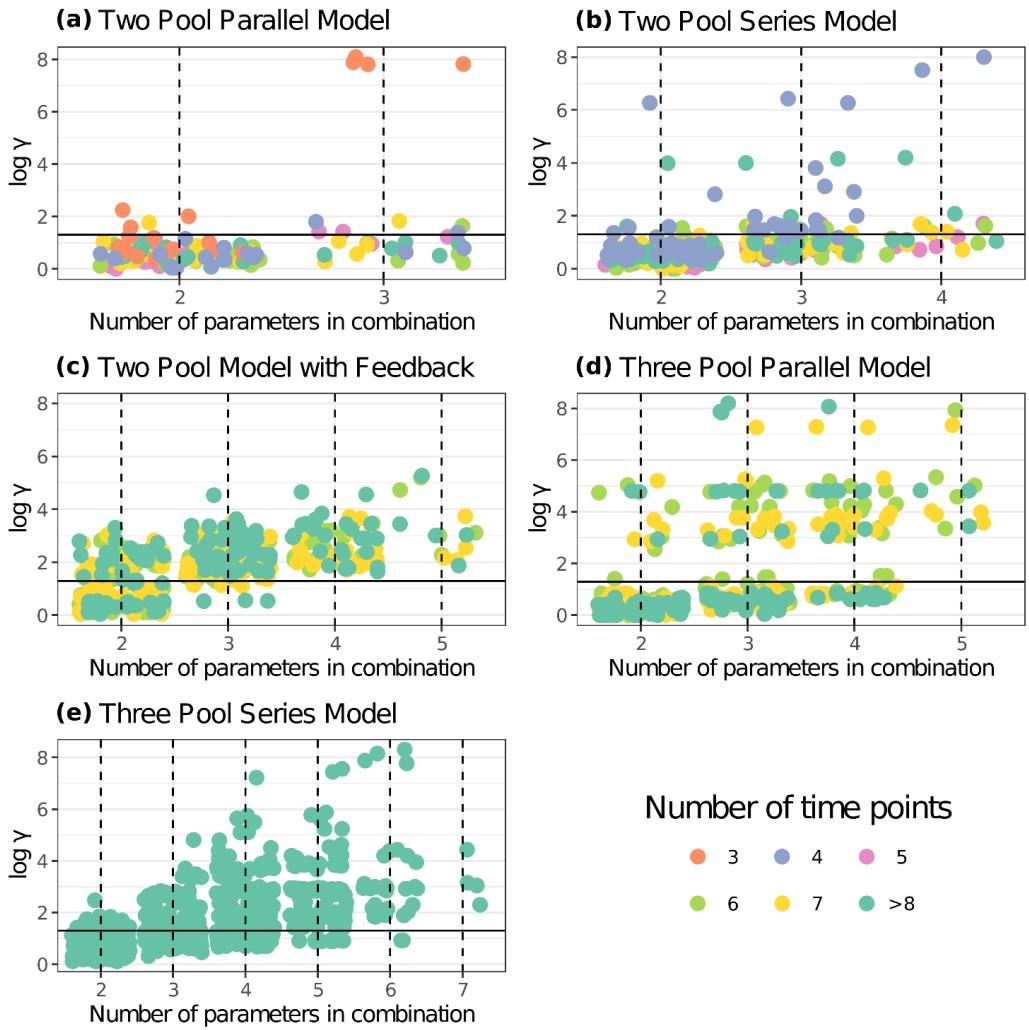

**Figure 5:** Collinearity index ($\gamma$) comparison for different model structures using entries from *aridec*. $\gamma$ values were $\log_{10}$ transformed and horizontal lines at $\log_{10}(20)$ denote the maximum value of $\gamma$ for a model to be considered identifiable. Infinite $\gamma$ values were not plotted. Each panel shows data for a different model structure. The number of entries from the database used for each model structure is reported in Table 3. Each point represents $\gamma$ for a model structure fitted for a specific data set with different parameter combinations. The color scale for data points shows the number of data points in each data set (i.e.: the number of sampling dates plus the initial date). Values with >8 time points were grouped for easier interpretation. The number of model variants fitted for each model structure and database entry were: n=30 for the two pool parallel model with all three parameters and n=90 with one restricted parameter (a); n=25 for the two pool series model with all four parameters, n=150 with two restricted parameters, and n=100 with one restricted parameter (b); n=15 for the two pool model with feedback and all five parameters, n=150 with two and three restricted parameters, and n=75 with one restricted parameter (c); n=15 for the three pool parallel model with all five parameters, n=150 with two and three restricted parameters, and n=75 with one restricted parameter (d); n=5 for the three pool series model with all 7 parameters, n=250 with five restricted parameters, n=300 with four restricted parameters, n=230 with three restricted parameters, n= 115 with two restricted parameters, and n=35 with one restricted parameter (e).

lower than 20. The analysis of the restricted model version with 2 estimated parameters generated 57.3 % of identifiable results out of 150 models.

When testing for the three-pool parallel model structure, we used one full model with five parameters and 24 model variations comprising from two to four parameters each. None of the full model data points showed collinearity index values lower than 20. Out of the restricted four parameter models only 34.7 % of them could be identifiable. When we specifically looked at the models where either $p_1$ or $p_2$ were fixed, none of them were identifiable. Restricted models with

three parameters produced a 68 % of identifiable results. Further, restricted models that only had $k$ values were not
identifiable. Finally, 89.3 % of models restricted to two parameters were identifiable according to our analysis.

The last model structure analyzed was the three-pool series structure which produced comparisons for models with all
seven parameters, plus 118 other model variants with different restricted parameters. None of the models with all seven
parameters were identifiable. Only 5.7 % of the models with six parameters were identifiable, none of which corresponded
to the models were either $p_1$ or $p_2$ were fixed. Models restricted to five parameters produced 10.5% of identifiable results.
Specifically looking at models with both fixed $p_1$ and $p_2$, 100 % of those were not identifiable. Restricting models to four
parameters generated 28 % of results with $\gamma > 20$. Models with three estimated parameters produced 54.9 % of identifiable
results. Lastly, 84.8 % of models restricted to only two parameters were identifiable.

Because two-pool parallel and series models with a fixed $p_1$ parameter showed the highest percentage of identifiable cases
in this first analysis, we did a second test with the whole database for these models and for their respective full-parameter
versions for comparison (Fig. 6). For the two-pool parallel model with the full set of parameters 58.7 % of entries were
identifiable, whereas restricting the $p_1$ parameter yielded 99.5 % of identifiable entries. For the more complex series
models the percentages of identifiable entries were much lower, with only 20.1 % for the restricted version and none
identifiable cases for the version with a full set of parameters. Clearly, restricting the number of parameters to be estimated
decreases collinearity, but the results are highly variable and dependent on each particular data set.

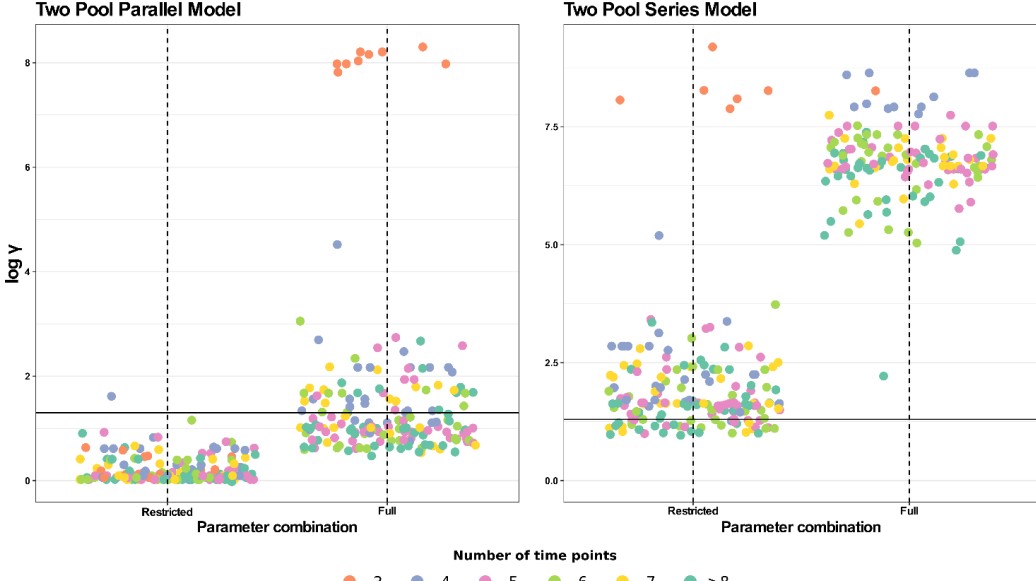

**Figure 6: Collinearity index (γ) comparison for four different model structures using the entire database. The full parameter
combination includes k₁, k₂, p₁ and a₁₂ (the latter only for the series model). The restricted parameter combination excludes
p1. γ values were log₁₀ transformed and horizontal lines at log₁₀(20) denote the maximum value of γ for a model to be considered
identifiable. Infinite γ values were not plotted. Each panel shows data for a different model structure. Each point represents γ
for a model structure fitted for a specific data set with different parameter combinations (n = 184, for each model structure).
The color scale for data points shows the number of data points in each data set (i.e.: the number of sampling dates plus the
initial date). Values with >8 time points were grouped for easier interpretation.**

### 3.3 Applied example

The previous analysis of collinearity index (Fig. 6) shows that from the proposed model structures to be fitted, our data
in most cases can be fitted to two pool parallel and series structure models with a restricted initial proportion of litter in
pool 1, aside from single pool models. Figure 7 shows the results from the simulation of the dynamics of organic matter
loss from leaf litter fitted from the *Day2018* entry. The one-pool model showed how a single reservoir of organic matter
from leaf litter decomposed at a $k$ rate of 0.0142 month$^{-1}$ (Fig. 7a). At the end of the almost three-year period the remaining
percentage of total organic matter was 61.02 %. The two-pool parallel model showed a fast-decomposing organic matter
pool with a $k_1$ of 0.0158 month$^{-1}$ (Fig. 7b). This pool went from representing 91 % of total organic matter to 52.4 % after
almost three years. On the other hand, the slow-decomposing pool, which we defined as the initial lignin content of litter,
had a value of $k_2$ of 1.5 x 10$^{-16}$ month$^{-1}$. We defined this pool as 9 %, and it remained unchanged after two years, which
was expected from a $k_2$ of nearly zero. Lastly, the two-pool series model showed a fast-decomposing pool with a $k_1$ of
0.11 month$^{-1}$ (Fig. 7c). This pool went from 91 % of total organic matter to 1.7 % at the end of the experiment. The slow-



decomposing pool in this case had a $k_2$ of 0.02 month$^{-1}$. This model also had a transfer coefficient from the fast-decomposing pool to the slow pool of 1 (i.e.: 100 % of organic matter in the fast pool that decomposed in a month transformed into more recalcitrant forms, adding to the slow-decomposing pool). Then, the slow decomposing pool went from 9 % at the start of the simulation to 54.08 % after almost three years. Judging by their AICc values, the three models were similarly supported by the data.


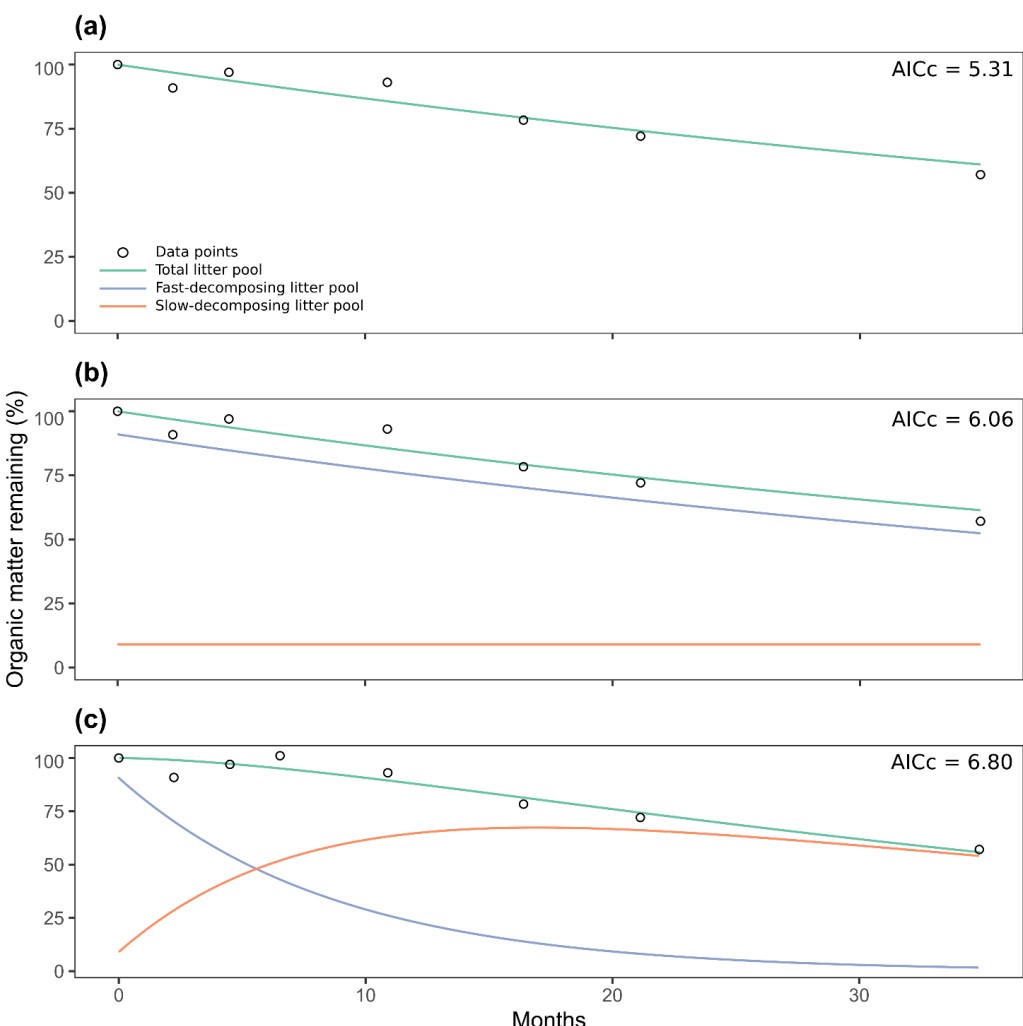

**Figure 7: Comparison of three different decomposition model structures fitted with time series of organic matter loss from the Day2018 entry: one pool model (a), two pool model with a parallel structure (b), two pool model with a series structure (c). AICc: Akaike Information Criteria, Bias Corrected.**

## 4    Discussion

### 4.1 The *aridec* database

The *aridec* database is a comprehensive database with a wide range of information on decomposition studies from aridlands worldwide, which includes litter mass loss data, litter traits and experimental design information. Our exhaustive bibliographic search gave us close to two hundred papers that fulfilled our criteria. Notably, we did not limit our work to





studies published in English: *aridec* includes also papers in Spanish, Portuguese, and Mandarin. This widens the scope of
       our work to achieve a more inclusive database.
       From a geographic perspective, study sites included in *aridec* cover most of the main aridlands of the world (Fig. 2). As
       expected, countries like China, USA, and Argentina, had the largest number of studies, which might be related to the
       extension of aridlands in these countries, since China has $6.07 \times 10^6$ km$^2$ of drylands (Huang et al., 2019), and around 40%
and 69% of USA and Argentina territories are considered as drylands, respectively (Verbist et al., 2010; White and
       Nackoney, 2003). In contrast, some of the biggest deserts in the world, such as the Sahara, the Kalahari, the Australian
       Outback, and the Arabian desert are underrepresented, if not absent, in our database. Future efforts should focus on this
       information void, and the *aridec* database will be available to include these coming studies in our framework.
       Study sites in the database represent a big part of the climate range where aridlands occur, from hyper arid deserts to dry
sub-humid ecosystems (Fig. 2). Some of the study sites (~12%) were classified as humid according to the GAI. We
       chose to include them because those studies reported marked dry seasons at the study sites and focused on seasonal patterns of
       litter decomposition. The range of climatic variables such as mean annual temperature and precipitation, physical
       variables like altitude, along with land cover types are also very well represented in this database (Fig. 4). This wide
       representativity of climates in *aridec* is a crucial asset if the database is to be used to answer global-scale questions.
Nonetheless, dry sub-humid lands but mainly hyper-arid deserts are the least represented in the database, suggesting more
       studies should be developed in these areas. Moreover, there is a void in the colder end of the climatic range of aridlands
       (Fig. 3). This might be related to the lack in *aridec* of sites of ecosystems like tundra where the GAI is mostly low, but a
       bibliographical search like ours could not retrieve the studies in those areas. There is clearly big room for expanding our
       database and increasing its potential applications.
One of the advantages of the *aridec* database is that its files are compatible with R and, specifically, with the *SoilR* library
       (Sierra et al., 2012). In addition to the fact that this database is an open-source and open-code project, there is huge
       potential for broadening the extent of the information in *aridec* and for developing code to work with it. Nonetheless we
       should mention that the formats included in the database are not R-exclusive and can be used with most commonly
       available software. YAML files can be read and edited from any text processor and CSV files can also be opened with
any spreadsheet software. Although statistical analysis on R scripts cannot be used elsewhere, raw data itself can be freely
       processed with any statistical software.
       The *aridec* database can be used on its own, but we recommend complementing our information with other publicly
       available databases to expand the application possibilities. Metadata are not always fully reported in publications, so it is
       possible to fill these gaps with climate and altitude data from databases like NASA Prediction of Worldwide Energy
Resources (POWER; NASA Langley Research Center (LaRC)). Leaf litter traits are a big part of the database that are
       sometimes poorly reported in publications. A general source of litter traits data can be obtained upon request from the
       TRY database (Kattge et al., 2020). Another more specific source of information is the NODdb where nodulating-N-fixing
       plant genera are detailed (Tedersoo et al., 2018).

       **4.2 Model fitting within *aridec***

Based on our collinearity analyses (Fig. 5 and 6), we suggest that before fitting any models to our data it is crucial to
       check the identifiability of the parameters with each database entry. We assume that all entries can be fitted to a one-pool
       model since there is only one parameter to estimate. As for the more complex models, the situation is highly dependent
       on which data entry is being used. For instance, although most entries could be used to fit two-pool models with parallel
       and series structures, there are some exceptions. Moreover, in most cases fitting these models is only possible by
restricting the parameters estimated to only decomposition constants and transfer coefficients. We suggest restricting
       models to these parameters because it is more likely to find data on initial proportion of lignin or cellulose to be used as
       proxies for parameters $p_1$ or $p_2$. Not only might they be more difficult to find in the literature but estimating values for
       decomposition constants and transfer coefficients might be altogether a better use of this database. Further, we recognize
       that for some specific combinations of parameters and data sets the more complex models might be identifiable (data
points below the $log_{10}(\gamma=20)$; Fig. 5). Besides collinearity, the number of degrees of freedom will restrict which models
       can be fitted to the data so these two aspects should be considered together. We provide an R script in the database to test
       collinearity for individual data sets.
       Again, interconnection between datasets like *aridec* and others like TRY (Kattge et al., 2020) is a key workaround to the
       collinearity problem by providing data for parameter restriction (Sierra et al., 2015). We recognize limitations in data
available from field studies ultimately limits our capability to model more complex models (Brun et al., 2001). Moving
       forward, new decomposition studies should consider making more measurements and including data on litter initial
       chemical quality. This will allow for the detection and modeling of finer-scaled dynamics of organic matter (see Appendix
       A).
       The applied example in Fig. 7 serves as a glimpse of the potential of this database. Choosing which model to fit with
decomposition data is not a minor question. For instance, none of the three models was supported as the best model from
       their AICc, which suggests they all must be considered when making conclusions from this simulation (Anderson and
       Burnham, 2004). Depending on our input data one could end up choosing different model structures from their AICc





values. Nonetheless, ecological theory may come into play here instead of applying purely statistical reasoning. As has been pointed out recently, litter decomposition is not as much a process of "what is lost" but more of "what is left" (Prescott and Vesterdal, 2021). A balance between statistical fit and theoretical support should be found when choosing which model is best for each study case.

The *aridec* database is available for open access and download at github.com/AgustinSarquis/aridec (Sarquis et al., 2022). Our hope is that newer studies on dryland litter decomposition will be added to the database by new collaborators (see Appendix A). It is important to follow our user guidelines to ensure consistency, all of which are available in the database itself. File templates for uploading new entries to the database are given and further details can be found in them. Additionally, users will find in the database a *README* file, scripts to test file consistency and many examples on how to apply functions and to fit models using R code.

To our knowledge, the *aridec* database is unique. Other databases that focus on land C studies include the Soil Incubation Database (SIDb; Schädel et al., 2020); a peatland productivity and decomposition parameter database compiled by Natural Resources Canada (Bona et al., 2018); and the Chilean Soil Organic Carbon Database (CHLSOC; Pfeiffer et al., 2020). Although they all intend to assess questions related to C budgets in terrestrial ecosystems to some extent, not all of them present decomposition data (i.e.: Pfeiffer et al., 2020). Moreover, only SIDb (Schädel et al., 2020) and *aridec* contain time series of organic matter loss. This is a unique asset that allows for future studies to make new assessments of decomposition without having to worry about inconsistencies in the calculation of $k$ parameters. Finally, none of these other databases is centered around plant litter decomposition in aridlands like *aridec*.

The extent of the information included in *aridec* in addition to its open-science approach makes it a great platform for future collaborative efforts in the field of aridland biogeochemistry. In this sense, the main purpose of this database is to further our understanding of C dynamics at the earth system level. Complete datasets like *aridec* are necessary to test which model structures and parameters best explain decomposition processes and to help develop more realistic representations of the global C cycle in drylands (Luo et al., 2016). A potential application of our database is to combine ecological data with climatic data in Earth System Models, which is a promising framework to assess future global change stresses and their effects on the biosphere (Bonan and Doney, 2018).

**APPENDIX A**

**Recommendations for future decomposition field studies**

Compiling published studies for the database led us to come up with a set of recommendations that scientists working on field decomposition studies may take into consideration if order to incorporate future entries in *aridec*.

- Coordinates: From the database, 7.6 % of entries had errors in their site coordinates and 8.7 % had no coordinates at all. That means that for 16.3 % of entries we had to either look for coordinates in other publications or search for the approximate location on Google Earth. Exact coordinates are a must for a study to be incorporated in geospatially explicit databases. Since nearly half of the problematic entries corresponded to typographic errors, we recommend authors and reviewers to check for the correctness of coordinates. Further, we suggest providing coordinates as exact as possible and to avoid using vaguely broad coordinates (e.g., reporting coordinates of the closest town to the study site).

- Soil classification: Out of all entries only 29.3 % reported soil taxonomy from the study site correctly. An additional 7.1 % of entries provided a classification for the soil, but they did not specify the classification system they used (i.e. FAO, WRB, or USDA). This is important because names of soil taxa are not always exclusive to a single classification system and their definitions are most unlikely interchangeable (Hughes et al., 2017). For the most part of studies this information might not be available, but for those where it is we suggest reporting it. Otherwise, making inference from soil types would be impossible.

- Mesh transmittance: Only 13.6 % of entries in *aridec* had measured the light transmittance of the mesh they used to construct litter bags. Light interception by mesh can be very high: as much as 50% of total radiation, photosynthetically active radiation, or ultra-violet radiation, as seen in our database. Considering the established importance of sunlight as a decomposition driver in aridland ecosystems (Austin et al., 2016), studies with mesh materials that block a significant proportion of light might be inducing unwanted effects, and underestimating effects of photodegradation. We recommend, if possible, choosing high-transmittance materials (the highest in our database has a 95 % transmittance of total radiation), measuring mesh transmittance and reporting these values in the manuscript.

- Sampling dates: The matter of choosing when to pick up samples from the field is complex. Ideally, the total amount of sampling dates and the amount of time between those dates should only depend on the hypothesis. The reality is that logistics has a huge impact on what scientists do, especially for field ecological studies. How researchers chose to set their sampling dates will determine the scale of the patterns that they will be able to detect from their experiments. For example, in some aridlands where decomposition is very slow, litter might take years to fully decompose, and short experiments are not able to capture this part of the process. Most of the studies in aridec lasted around a year, with only



a few studies lasting longer (Fig. 4b). Further, in some systems leaching can have a big impact during the first days to weeks of decomposition, and more frequent sampling at the beginning of the experiment may allow to detect this. In our database, most studies made measurements less than once a month (Fig. 4d), meaning that only monthly to yearly processes could be detected. These limitations extrapolate to modelling challenges: it is not possible to fit data to models that represent patterns that went undetected due to the study design. To accurately estimate decomposition rates ($k$), it is thought that litter at the last sampling date needs to have lost at least 50% of mass. As such, this suggests that the number of samplings and extension in time of the study should reflect these goals. We suggest that researchers are aware of all these issues, and that also they have enough pickups to actually be able to calculate slopes of the relationships, which increases enormously the power of inference.

- Corrections of mass loss measurements: After collection from the field, samples in most cases carry with them moisture and inorganic matter from the site. This of course can underestimate measurements of litter mass loss. Once in the laboratory, samples should be cleaned of any extraneous material and their moisture content measured. After this, a portion of each sample should be used for quantifying the proportion of ashes (Harmon et al., 1999). This should be done as well for samples that were not taken to the field and are used for measurements of initial litter traits. All mass loss analysis should be done on an oven-dried ash-free basis.

**-** Time series: A large number of studies could not be included in *aridec* because they only published decomposition rates. As much as this is a common practice, it limits the possibilities for incorporation into databases like ours and for further analysis that might need temporal dynamics data as input. We suggest not only providing averaged values of mass loss over time, but also raw data as supplemental material. This helps bridge the reproducibility gap in ecological studies and represents a step forward to an Open Science approach (Hampton et al., 2015).

- Initial litter quality: The characterization of litter chemical and physical traits at the beginning of experiments is an important tool for answering research-specific questions of decomposition studies. However, from our results it was evident that these initial litter traits are also useful to decrease model collinearity (Fig. 6). Particularly, the initial content of litter components that constitute a big part of total mass like cellulose, acid detergent lignin or water-soluble sugars can be used as proxies for the initial proportion of litter mass in pools of different decomposition rates. Unfortunately, not even half of the studies in *aridec* reported initial lignin content for each litter type. We managed to complete up to 48 % of lignin content data by averaging across database entries of the same species and by requesting data from TRY database. To our surprise, we only found lignin values for 3 out of 236 litter types that we searched in TRY database. This leads us to suggest that not only should authors measure and report these characteristics of interest, but they should also contribute their data to open access databases from which other scientists can benefit.

**Data availability**

Version 1.0.1 of *aridec* is publicly available at https://doi.org/10.5281/zenodo.6025969 (Sarquis et al., 2022). Documentation of the project and the R package are presented on the project's website (https://github.com/AgustinSarquis/aridec, last access: 9 February 2022). The database is open for reuse, and the usage license follows the GPL-3 license (https://opensource.org/licenses/GPL-3.0, last access: 9 February 2022). When using the database or R package, users should cite this definition publication and consider citing individual studies (publication or dataset).

**Code availability**

All scripts necessary to obtain figures in this publication are included in the database inside the "scripts" folder.

**Conclusions**

The *aridec* database is a comprehensive database with a wide range of information on decomposition studies from aridlands worldwide. Study sites included in *aridec* cover most of the main aridlands of the world and represent well the range of climatic conditions that characterize aridlands. We found that although many studies have been conducted in arid lands, there is low representativity in cold arid regions, where new studies should be performed to obtain a more comprehensive understanding of decomposition in arid lands worldwide.
Our identifiability analysis showed that the information content in litter decomposition studies can only inform simple models with one or two pools. More complex models can be obtained for datasets with multiple data points, and a well characterized initial litter mass quality (such as lignin or cellulose content), which will result in low collinearity index values and will allow for enough degrees of freedom.



580 One of the best assets of the *aridec* database is that its files are compatible with R and the SoilR package, making collaborative work more direct and approachable. Although our application suggestions are based on the use of the SoilR package, we recognize that other approaches might be suitable for the use of this database.
To our knowledge, the *aridec* database is unique and the extent of the information included here in addition to its open-science approach makes it a great platform for future collaborative efforts in the field of aridland biogeochemistry.

**Authors contributions:**

585 ATA and CS conceptualized this project. AS and IS curated the data. AS and CS created the methodology. AS analyzed the data. ATA and CS acquired funding for this project. CS supervised this project. AS wrote the original draft of this manuscript. CS, ATA, and IS reviewed and edited the manuscript.

**Competing interests**

590 The authors declare that they have no conflict of interest.

**Acknowledgments**

C. Casas, M. Jardón and N. Moreno for making a revision of an early version of the manuscript.

**Financial support**

Financial support for the project came from the University of Buenos Aires (UBACyT 2020), the Agencia Nacional de
595 Promoción Científica y Tecnológica (ANPCyT; projects PICT 2015-1231, PICT 2016-1780, and PICT 2019-02645). AS was funded by the University of Buenos Aires (UBACyT 2018; Res. Nº 1245/18) and the German Academic Exchange Service (DAAD; Research Grants - Short-Term Grants program 2021, ID: 57552337).

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
