# Peer review of "aridec: an open database of litter mass loss from aridlands worldwide with recommendations on suitable model applications"

_Earth System Science Data, 2022_

## Author Response (AR1)

**Author's response**

**Manuscript:** essd-2022-54

**Referee number 1:**

Comment 1: In the entry of Almagro et al.,2017, "organic matter remaining%" is shown in the database, but it's "Ash-free dry mass remaining%" in the article;

Answer: We considered ash-free dry mass and organic matter remaining as equal in the database. This is because the ash-free dry mass correction assumes that ash is inorganic matter and thus ash-free mass is equivalent to organic matter for the purposes of these studies. Although we can see that this might generate some confusion, we did not change this in the database entry. Nevertheless we made the following changes in lines 172-175: "Litter mass loss units should be expressed either in (dry) mass remaining, organic matter remaining, or C remaining. Organic matter remaining is in our database a synonym of ash-free dry mass remaining. This is because the ash-free dry mass correction assumes that ash is inorganic matter and thus ash-free mass is equivalent to organic matter for the purpose of this database (Harmon et al. 1999).".

Comment 2: In the entry of Tian et al., 2007, "organic matter remaining%" is shown in the database, but it's "mass remaining" in the article;

Answer: Although in the original figure they report "mass remaining", in Methods the authors clarified that they did an ash-free correction (page 220). As such, similar to Comment 1, we considered this to be organic matter remaining and did not change the entry.

Comment 3: In the entry of DiedhiouSall et al., 2013, the litter initial C/N ratio value seems to be inconformity with the article;

Answer: Thank you for pointing out this incongruity, which we observed as well. The C:N data in the article is inexplicably high and does not coincide with the calculation of C to N values reported in the manuscript. Thus, we decided to do this division ourselves and realized there must be a typographic error in the article. All values in the article are approximately 10 times higher than the calculated values, so we divided them by 10 and used these values instead. We clarified this in the entryNote of the metadata.yaml file: "C:N data must be wrong in the paper, we divided values by 10 in initConditions.csv".

Comment 4: In the entry of Levi et al., 2020, "starting month" is January in the database, but it's July in the article;

Answer: Thank you for this observation. We corrected this accordingly in the entry.

Comment 5: Figure 2, it will be better to show the distinction of sites by ecosystem type;

Answer: Thank you for this suggestion. We included this modification in Figure 2 and added the following in the figure description: "Points represent study sites in the aridec database. Colors represent different ecosystems as reported in the original publications".

Comment 6: Line 319,"According to the GAI()...", delete the " ()".

Answer: We corrected this accordingly in the manuscript.

Comment 7: Figure 4.(a)(b), the ordinate axis title "number of pubblication", should be " number of publication ".

Answer: We corrected this accordingly in Figure 4.

Comment 8: Figure 7, there are 8 data points in (c), but 7 data points in (a) and (b). For a representation of the same data set for comparison, please check the raw data.

Answer: Thank you very much for this observation. There was one missing point in figures 7a and b. We corrected this accordingly.

Comment 9: In discussion section 4.2, it would be useful to summarize the results of model fitting/model selection for different ecosystem types or GAI classification groups.

Answer: Thank you for this suggestion, which inspired the following modifications in lines 476-485: "Again, interconnection between datasets like *aridec* and others like TRY (Kattge et al., 2020) is a key workaround to the collinearity problem by providing data for parameter restriction (Sierra et al., 2015). We recognize limitations in data available from field studies ultimately restrict our capability to fit more complex models (Brun et al., 2001). This limitation can have further implications if we consider the proportion of identifiable data sets per ecosystem type or level of aridity. For example, semi-arid and dry sub-humid ecosystems show the lowest proportion of identifiable data sets for a two-pool parallel model with all parameters (data not shown). This would lead to an under-representation of some aridlands because of a lack of suitable data available. Moving forward, new decomposition studies should consider making more measurements and including data on litter initial chemical quality, as well as expanding studies to less represented climates and ecosystems. This will allow for the detection and modeling of finer-scaled dynamics of organic matter (see Appendix A)."

**Referee number 2:**

Comment 10: The authors argued "a central application of this database is the development of models of litter decomposition for aridlands", and they used a subset of the database (30 entries) to examine the identifiability of various model structures (1-, 2-, 3-pool models, with parallel and series structure for last two). The results show that two-pool parallel and series models with one fixed parameter have the highest identifiable percentage. I'd like to learn more opinions from the authors on whether and how the data samples could affect this result. In these 30 entries, about half of them have more than 5 time points, and only five entries have more than 8 time points. Will including more sites with >8 time points change the result? For the

three-pool model, it seems like the series structure has a better identifiability under 2 parameters (Fig 5d and 5e). Is it possibly caused by the entries used (only sites with >8 time points used in three pool series)? If these models are tested by different subsets of the database, how do you justify the identifiability comparison among them?

Answer: Thank you for this comment, and for bringing up the issue of sample size or number of time points in this type of datasets. As can be seen from Figure 4c, the number of time series with more than 8 time points is relatively small. For this comparison we decided to include 5 entries for each time length, but for >8 we had to pool entries with different time lengths together for better representation. Generally, the larger the number of time points, the lower the collinearity among parameters. But this is highly dependent on each dataset, as can be seen in Figure 5b where one dataset of >8 points is not identifiable using all four parameters. As models get more complex, we had to progressively exclude entries with less time points because they did not have enough degrees of freedom. Because of this, it would be impossible to compare the collinearity for all models fit with data sets of all time lengths. Thus, what we can compare in this Figure is how entries with equal number of time points behave under each model structure. We had already included a short explanation for this in the Methods section (lines 286-289): "For the identifiability analysis, we first selected a representative group of 30 entries from the database ranging from 3 to 19 time points (Table 2). The number of data points in time limits the number of parameters that can be fitted because it affects the number of degrees of freedom. Thus, models with more parameters require longer data sets. This meant that, a priori, not all entries could be used to fit all model structures." We had also mentioned this in the Discussion section (lines 464-465): "Besides collinearity, the number of degrees of freedom will restrict which models can be fitted to the data so these two aspects should be considered together". To further clarify this, we added the following lines in the manuscript: "Here we can compare how entries with equal number of time points behave under each model structure." (lines 346-347); "For instance, although most entries could be used to fit two-pool models with parallel and series structures, there are some exceptions, and, as can be seen in Figure 5b, one dataset of more than eight points was not identifiable for a model with all four parameters." (lines 462-464); and "As models get more complex, we had to progressively exclude entries with less time points because they did not have enough degrees of freedom" (lines 467-467).

Comment 11: Same question for Figure 6: whether the same or a different number of entries were used for testing these models? If not, the results revealed are not as much as percentage change by using the same denominator (entry numbers), but more like the model collinearity for certain groups of sites. I am not convinced this is an apple-to-apple comparison.

Answer: In Figure 6, we used the entire database. As such, it would not have been possible to use other data. Of course, if we did this analysis with a completely different database the results would likely be different as well. The results in Figures 5 and 6 are not to be interpreted as how those models perform in general, but how they perform with the specific data in *aridec*. And still, collinearity can vary a lot for entries with the same number of time points. That is why we recommend users to test collinearity before fitting these or other models. We made the following modification in lines 467-469: ". As a concluding remark, the results in Figures 5 and 6 are not to be interpreted as how those models perform in general, but how they perform with the specific data in *aridec*. That is why we provide an R script in the database to test collinearity for individual data sets". Thank you for your comment.

Comment 12: in the following applied case study (Figure 7), why do you only select Day2018? Will the data entry with more time points be able to separate the performance of different model structures? In Fig 7a and 7b, only seven observed time points were included. I am curious why the observed litter mass first declined and then increased to a level close to T0. Is this dynamic caused by more litter input?

Answer: In order to fit all three models we needed to find an entry for which models were identifiable. And, as a consequence, we could only fit the two-pool series model if the authors had provided additional data to restrict the mass in pool 1. As we mentioned in Methods (lines 305-311): "Restricting the p1 parameter is a sensible way of achieving identifiability because it is common to find information on litter lignin content in decomposition publications, and this can be used as an initial proportion value for the slow-decomposing litter pool (i.e.: p2). Since

$$p1 + p2 = 1, \hspace{6cm} (7)$$

then it is possible to estimate the p1 as the complementary value of p2." Not all entries in the database have lignin data (see Appendix A – Initial litter quality), so this further limited which entries we could use for this demonstration. And, since Figure 7 serves only as a demonstration, we did not intend to use more entries. One could expect to separate the performance of different model structures, as you say, but not necessarily with a larger number of time points. Whether a time series has a better fit with one model or another depends mostly on mass loss dynamics that were captured by the experimental design, in our opinion. Yes, longer datasets can be fitted with more complex models, but if the overall mass loss dynamics fit better for a one-pool model, higher model complexity will not be chosen using AIC. As for the different number of time points in Figure 7, this was a graphic mistake and we corrected it accordingly. Lastly, this apparent increase in mass over time after the third sampling date was not addressed in the original paper, but it has been observed in many studies. This is usually attributed to the growth of microbial biomass on litter samples. We made the following modification in lines 489-491: "Whether a time series has a better fit with one model or another depends mostly on mass loss dynamics that were captured by the experimental design. While longer datasets can generally be fitted with more complex models, if the overall mass loss dynamics fit better for a one-pool model, higher model complexity will not be chosen using AIC."

Comment 13: The applied example is very simple and doesn't show data capability in different models. I doubt the reuse potential unless there are more data with multiple time points (this information seems missing or unclear for the entire database, I only find the entry percentage of time points in the 30 entries). This manuscript could be improved if the authors provide further investigation on 1) how aridland litter mass loss measurement can be improved to better inform modeling development, and/or 2) what modeling efforts can be targeted to change from a data perspective (e.g., missing or underrepresented mechanisms, behaviors of the two-pool model with parallel, series, and feedback structures, etc.).

Answer: Thank you for your comment. First, the number of time points, sampling dates or harvests over time for the entire database are shown in Figure 4c. As you can see, most studies have 5 time points while only a few have longer time series. Second, we outlined a number of suggestions on how future studies can be improved to increase our modelling capabilities in Appendix A (lines 520-578). Lastly, we modified lines 510-519 as follows: "The extent of the information included in aridec in addition to its open-science approach makes it a great platform

for future collaborative efforts in the field of aridland biogeochemistry. In this sense, the main purpose of this database is to further our understanding of C dynamics at the earth system level. Complete datasets like *aridec* are necessary to test which model structures and parameters best explain decomposition processes and to help develop more realistic representations of the global C cycle in drylands (Luo et al., 2016). Further, additional parameters could be used to test the importance of mechanisms that are relevant in aridlands, but are underrepresented in the literature. Studies on processes like photodegradation (Adair et al., 2017) could be expanded to a wider geographical range and to soil processes thanks to the representation of sites in aridec using the SoilR framework (Sierra et al., 2012). Another potential application of our database is to combine ecological data with climatic data in Earth System Models, which is a promising framework to assess future global change stresses and their effects on the biosphere (Bonan and Doney, 2018)."